# Identification and *in Silico* Characterization of Novel and Conserved MicroRNAs in Methyl Jasmonate-Stimulated Scots Pine (*Pinus sylvestris* L.) Needles

**Baiba Krivmane** [1], **Ilze Šņepste** [1], **Vilnis Šķipars** [1], **Igor Yakovlev** [2] , **Carl Gunnar Fossdal** [2], **Adam Vivian-Smith** [2] **and Dainis Runģis** [1,*]

[1] Genetic Resource Centre, Latvian State Forest Research Institute "Silava", Riga street 111, Salaspils LV-2169, Latvia; baiba.krivmane@silava.lv (B.K.); ilze.snepste@silava.lv (I.Š.); vilnis.skipars@silava.lv (V.Š.)
[2] Norwegian Institute of Bioeconomy Research, Postboks 115, Ås NO-1431, Norway; igor.yakovlev@nibio.no (I.Y.); Carl.Gunnar.Fossdal@nibio.no (C.G.F.); adam.vivian-smith@nibio.no (A.V.-S.)
[*] Correspondence: dainis.rungis@silava.lv; Tel.: +371-2834-4201

**Abstract:** MicroRNAs (miRNAs) are non-protein coding RNAs of ~20–24 nucleotides in length that play an important role in many biological and metabolic processes, including the regulation of gene expression, plant growth and developmental processes, as well as responses to stress and pathogens. The aim of this study was to identify and characterize novel and conserved microRNAs expressed in methyl jasmonate-treated Scots pine needles. In addition, potential precursor sequences and target genes of the identified miRNAs were determined by alignment to the *Pinus* unigene set. Potential precursor sequences were identified using the miRAtool, conserved miRNA precursors were also tested for the ability to form the required stem-loop structure, and the minimal folding free energy indexes were calculated. By comparison with miRBase, 4975 annotated sequences were identified and assigned to 173 miRNA groups, belonging to a total of 60 conserved miRNA families. A total of 1029 potential novel miRNAs, grouped into 34 families were found, and 46 predicted precursor sequences were identified. A total of 136 potential target genes targeted by 28 families were identified. The majority of previously reported highly conserved plant miRNAs were identified in this study, as well as some conserved miRNAs previously reported to be monocot specific. No conserved dicot-specific miRNAs were identified. A number of potential gymnosperm or conifer specific miRNAs were found, shared among a range of conifer species.

**Keywords:** microRNA; isomiR; Scots pine; IonTorrentPGM; methyl jasmonate; precursor microRNA

## 1. Introduction

Scots pine is a long-lived organism with a wide distribution in the northern hemisphere. This requires an adaption to a broad range of growing and environmental conditions—likely facilitated by substantial phenotypic plasticity and epigenetic variation [1]. Plants have developed a range of epigenetic mechanisms to deal with biotic and abiotic stress, including DNA methylation, histone modification and expression of various non-coding RNAs (including microRNAs), which influence gene expression and regulation [2]. MicroRNAs (miRNAs) are a class of non-protein coding small RNAs (sRNAs) of ~20–24 nucleotides in length that play an important role in a variety of biological and metabolic processes, primarily through the coordinated action on the post-transcriptional control of gene expression [3]. While miRNAs control essential aspects of plant growth and development, miRNA and sRNA expression are similarly important in the responses to the challenges of stress and pathogens [4–7]. In plants, miRNA studies have been mainly concentrated in angiosperms and relatively few reports have been made in conifers.

MiRNAs are initially synthesized as primary miRNA transcripts (pri-RNAs), which can be up to several hundred nucleotides in length and contain at least one hairpin stem loop structure. A range of enzymes process these transcripts (DICER-LIKE 1 enzyme, HASTY and others) to generate miRNA precursors (pre-miRNAs) and subsequently produce the mature miRNA molecules [5,8]. In addition, alternative miRNA processing pathways have been described in plant species that involve DICER independent pathways and the processing of other non-coding RNAs [9]. These include miRNAs that are produced from an intron of a protein-coding gene by DICER-independent pre-mRNA splicing machinery (miRtrons), Argonaute RISC catalytic subunit 2 (AGO2) processing, and origins from various other noncoding RNA genes. Plants produce many distinct types of DCL/AGO-associated regulatory sRNAs, from which miRNAs, phased siRNAs and heterochromatic siRNAs are three of the major types of small plant RNAs [10,11].

miRNA precursors can produce both precise excision or distinct miRNA variants known as isoforms (isomiRs). IsomiRs from the same miRNA arm typically differ at their 5′ end, 3′ end, or both, thus abundant isoforms are frequently annotated alongside a canonical miRNA sequence found in databases. IsomiRs in plants can originate from imprecise cleavage by DCL1 (templated isomiRs), which generates variants that show complementarity to their pre-miRNA. Alternatively, isomiRs can be spawned by post-transcriptional modification due to the addition or removal of specific nucleotides to miRNA ends (non-templated isomiRs) [12]. Non-templated modifications most frequently occur at the 3′ end, but less frequently at the 5′ end, and these modifications can influence both the miRNA stability and the efficiency of target repression) [13,14]. One of the mechanisms that can increase the diversity of miRNA action is seed shifting, where the dominant mature miRNA isoform is shifted in complementarity relative to its target by one to several nucleotides in the 5′ or 3′ direction relative to its original position. Therefore, modifications at the 5′ end of the mature product could relocate the seed position and thus change in the seed sequence (called "seed shifting"), thereby altering mRNA target recognition and function [15].

While there are relatively few conifer sequences deposited in specific miRNA databases (e.g., miRBase), there have been a number of reports published about miRNA studies in conifer species. These species include lodgepole pine (*Pinus contorta*) [16], Eastern white pine (*Pinus strobus*) [17], larch (*Larix leptolepis*) [18–22], Norway spruce (*Picea abies*) [23–26], Chinese fir (*Cunninghamia lanceolata*) [27,28], Chinese yew (*Taxus chinensis*) [29], sequoia (*Sequoia sempervirens*) [30] and others. However, there are no publications to date about miRNA studies in Scots pine (*Pinus sylvestris* L). The majority of miRNA studies in gymnosperms have investigated various developmental stages or plant tissues, e.g., the expression of small RNAs (sRNAs) in *Sequoia sempervirens* during phase changes, specifically in the juvenile, adult and *in vitro* propagated plants [30], expression patterns of conserved miRNAs from mature and germinated pollen of *Pinus taeda* [31], and miRNAs in zygotic embryos and female gametophytes of *Pinus taeda* [32]. Yakovlev et al. [26] reported on the expression of miRNAs in Norway spruce seedlings derived from plants regenerated after embryogenesis in a cold and warm environment. In addition to developmental processes, miRNAs are involved in regulation of defense responses. Differentially expressed miRNAs target *NB-LRR* genes in the bark of Norway spruce are produced in response to inoculation with *Ceratocystis polonica* and wounding [24], while in *Pinus taeda* miRNAs were also identified in the xylem during the process of fusiform rust gall development [33].

miRNAs are known to modulate the expression of genes during plant defense and the effect of methyl jasmonate on miRNA expression has also been investigated. The action of miRNAs on upon the biosynthesis and perception pathway is of particular importance in trees since jasmonic acid (JA) and its methyl ester, methyl jasmonate (MeJ), are plant signaling molecules that broadly affect gene expression impacting both plant growth and development as well as the response to pathogen attack, wounding and plant responses to abiotic and biotic stresses [34,35]. MeJ, which is synthesized from linolenic acid, and is one of the few plant compounds that are effective at low vapour concentrations. Jasmonates include jasmonic acid, its derivatives and conjugates; the jasmonates and in particular, the active hormone jasmonoyl-isoleucine is known to regulate defenses against

chewing herbivores [36]. Remarkably MeJ has been reported to be involved in defense priming, and the induction of anatomical and chemical changes such as the formation of traumatic resin ducts in the xylem, and the synthesis of phenolic and alkaloid compounds in many conifer species [37–39]. The interaction between miRNA-controlled gene expression and MeJ biosynthesis and perception is less known in gymnosperm species, but in *Taxus chinensis* cells, marked changes in miRNA profiles were reported in response to MeJ [29]. Likewise, the mechanisms by which MeJA regulates paclitaxel biosynthesis were also investigated in *Taxus × media* cells, and this study showed the potential for mRNAs being targeted by miRNAs [40].

The aim of this study was to identify and characterize novel and conserved miRNAs expressed in MeJ-treated Scots pine needles. In addition, potential precursor sequences and target genes of the identified miRNAs were determined, to understand the type of processes regulated by both conserved and novel miRNAs under stress conditions. A combined strategy—high throughput sequencing and computational prediction—was utilized to identify conserved *P. sylvestris* miRNAs from six sRNA libraries. The obtained mature miRNA sequences were analyzed and filtered based on known characteristics of plant miRNAs, and compared to other plant miRNAs available in databases. Conserved miRNAs are identified and their role is discussed.

## 2. Materials and Methods

Six one-year-old Scots pine ramets of one clone derived from Latvian Scots pine breeding program (Sm9-III-2) were transferred into growth chambers two weeks prior to the start of the experiment. Three of the ramets were each treated with 5 ml of a 10 mM MeJA/0.1% Tween 80 solution in deionized water, applied with a hand sprayer. Three control ramets were treated with 5 ml of a 0.1% Tween 80 solution in deionized water. The MeJA treated and control ramets were kept in separate growth chambers. After treatment, ramets were covered with plastic bags for 48 hours to allow the volatilization of excess MeJA. Ramets were grown at 17–22 °C under long day conditions (16 h light + 8 h dark). Two weeks after MeJA treatment, needles were collected and immediately frozen in liquid nitrogen. Control samples were harvested from untreated ramets in the separate growth chamber at the same time. Needle samples were stored at −80 °C until RNA extraction. Total RNA was isolated from 100 mg of needles using a phenol: chloroform: isoamyl alcohol extraction protocol [41]. Total RNA and small RNA quality, quantity and integrity number (for total RNA) was verified using the Agilent Technologies 2100 Bioanalyzer with RNA Agilent RNA 6000 Nano Kit and Agilent Small RNA kit. Total RNA preparations were stored at −80 °C. Total RNA samples were enriched for small RNA as outlined in the Ion RNA-Seq Library Preparation guide (Thermofisher Scientific Manual 4476286 revision E) and 6 small RNA non-barcoded libraries were prepared using Ion Total RNA-Seq Kit v2 for Small RNA Libraries (Part No. 4476289, Thermofisher Scientific) according to the manufacturer's protocol. Each amplified sRNA library was quantified and the quality analyzed using the Agilent Technologies 2100 Bioanalyzer with a High Sensitivity DNA Kit as intended for templating and separate sequencing as individual non-barcoded libraries on separate sequencing chips. Template-positive Ion Sphere™ Particles (ISPs) were prepared with the Ion OneTouch™ 2 Instrument and enriched with the Ion OneTouch™ ES following the manufacturer's protocol. The ISP enrichment was then assessed using the Qubit®2.0 Fluorometer and Ion Sphere™ Quality Control Kitl. Enriched ISPs were then loaded onto an Ion 316 chip (Cat. No. 4483188) and sequenced on an Ion Personal Genome Machine®(PGM™) System at the Norwegian Institute of Bioeconomy Research (NIBIO). The sequences were base called on the Ion Torrent Server with version 4.0.2.

The sRNA sequences were analyzed using the CLC Genomics Workbench software version 7.5.1 (QIAGEN). Low quality reads and adapter sequences were removed and sequences were filtered by length for miRNA identification: minimum length 19 nt and maximum length 25 nt [42,43]. As is known, plant miRNAs tend to be 21 or 22 nt in length and, as previously reported by Axtell and Meyers [11], no RNAs < 20 nt or > 24 nt should be annotated as miRNAs, and annotations of 23-or 24-nt miRNAs require extremely strong evidence; then, cut off limits were used from 19–25 nt. Conserved

miRNAs were identified by comparison with miRNA sequences from various tree species (*Pinus taeda*, *Pinus densata*, *Picea abies*, *Populus trichocarpa*, *Populus euphratica*, *Acacia auriculiformis*) as well as other plant species (*Arabidopsis thaliana*, *Oryza sativa*, *Nicotiana tabacum*, *Vitis vinifera*, *Zea mays*). Mature miRNA and pre-miRNA sequences of these species were obtained from miRBase (v20 and 21) [44,45]. Two mismatches were allowed between Scots pine miRNA sequences and miRNAs obtained from miRBase. Using the CLC genomics Workbench software, sequences were counted and assigned to families by comparison with mature miRNA sequences from miRBase. The parameters for sequence comparison were: additional downstream bases-2, additional upstream bases-2, missing downstream bases-2, missing upstream bases-2. miRNA sequences obtained from published reports, which were not present in miRBase, were compared to sequences in miRBase as well as sequences obtained in this study. The family classification utilised in miRBase was used to categorize miRNAs not published in miRBase. miRNAs and their isoforms, that contained more than 2 mismatches with published miRNAs were considered as potential new miRNAs for further validation.

In an attempt to verify both the conserved and novel mature miRNAs sequences obtained in this study, potential precursor and target gene sequences were identified. To identify potential Scots pine precursor miRNA sequences, both the conserved as well as unannotated miRNA sequences were aligned to the *Pinus* PGI_v9.0_032811 unigene sequences (available from DFCI Pine Gene Index at ftp://occams.dfci.harvard.edu/pub/bio/tgi/data/Pinus/). The miRA tool [46] was used for identification of pre-miRNAs. Small RNA sequences were mapped to *Pinus* unigene sequences (allowing two nucleotide mismatches) using the CLC genomics workbench software, and the miRA tool was used for identification of canonical miRNAs, 5′ and 3′ isomiRs and polymorphic isomiRs and precursor sequences. Previously described criteria [46] were used with some modifications: minimum length (in nt) of the double stranded segment within the folded sequence-19; minimum length (inclusive, in nt) of mature/star miRNA-19; maximum length (exclusive, in nt) of mature/star miRNA-25.

In addition, for conserved miRNA sequences, the minimum free-folding energy index (MFEI) [47] was calculated to confirm that the precursor sequences conformed to the requirements for forming the miRNA precursor structures [11]. Sequences with a maximum of two mismatches with the miRNA sequences were identified and taking into account reports that plant pre-miRNAs vary from approximately 80–200 nt in length [47], regions flanking the mapped mature miRNAs (approximately 150 nt downstream and 150 nt upstream) were used to predict folding structures using the Mfold program (http://mfold.rna.albany.edu/?q=mfold) web server [48] and the CLC genomics workbench software. If the length of a sequence was less than 300 nt, the entire available sequence was used as a miRNA precursor sequence. MFE (minimal negative folding free energy, $\Delta G$), AMFE (adjusted MFE), MFEI (minimal folding free energy index), length of sequence, nucleotide percentage (A, U, G, and C), A + U content, G + C content, and number of base pairs were calculated [49]. A sRNA sequence was considered as a potential miRNA candidate only if it met the following criteria [10,11,47,50]: (1) the predicted mature miRNA was allowed to have 0–2 nucleotide mismatches with the best matched known plant mature miRNA and sequence length was between 19 and 25 nucleotides; (2) a RNA sequence could fold into an appropriate stem–loop hairpin secondary structure; (3) the predicted mature miRNA is located on one arm of the hairpin structure; (4) there were less than 6 mismatches in the complementary site (the opposite miRNA* sequence on the other arm); (5) three mismatched positions of nucleotides in asymmetric bulges; (6) the predicted pre-miRNA had a high negative minimal free– folding energy (MFE) from which the negative folding free energies and MFE index (minimal free folding energy index, MFEI) were calculated in order to distinguish potential precursor miRNAs from other small RNAs. The MFEI was calculated using the formula:

$$[(\text{MFE}/\text{length of the RNA sequence}) *100]/(\text{G+C}) \% \qquad (1)$$

Predicted secondary structures of precursor miRNAs have folding free energy indexes (MFEIs) >= 0.85, distinguishing them from other small RNAs such as tRNAs, and rRNAs whose MFEI are

between 0.59 and 0.66; 7) 30–70% A + U content [47]. Application of these criteria can significantly reduce false positive identification of potential precursor miRNAs [50].

Identification of potential miRNA target genes was done by searching for complementary regions between the identified miRNAs in this study and by using all the *Pinus* transcript sequence input using online web server, and the psRNATarget-Plant Small RNA Target Analysis Server as described previously [51]. Potential target genes were annotated according to GO categories using the Blast2GO PRO Plugin and all non-redundant GeneBank CDS translations + PDB + SwissProt sequences as well SwissProt –non-redundant UniProtKB/SwissProt sequences [52].

## 3. Results

### 3.1. Sequencing of Scots Pine Small RNA Libraries

Sequencing of the six small RNA libraries yielded approximately 5.8 million reads before trimming. Prior to trimming, the average length of small RNA reads in the control sample libraries was 21.57 nt and 20.76 nt in the MeJA treated sample libraries. After trimming (19–25 nt), 4.5 million reads remained, with an average length 21.50 nt in the control libraries and 21.46 nt MeJA treated libraries. Small RNAs of 21–22 nt length were the most prevalent among the obtained sequences. In total, 1,021,696 unique small RNA sequences were found.

### 3.2. Identification of Conserved and Novel miRNAs in Scots Pine

To identify conserved miRNAs expressed in Scots pine, all unique small RNA sequences were compared to annotated mature plant miRNAs in miRBase. Sequences from 11 species in miRBase were utilised, of which six are woody species, including three conifer species-*Pinus taeda*, *Pinus densata*, *Picea abies*. Of the 1,021,696 unique small RNA sequences obtained, 4975 potentially conserved miRNA sequences were identified (consisting of 317,195 reads from a total of 4,488,459). Of these, 957 were ambiguously annotated, i.e., a small RNA sequence was similar to the mature regions of two different miRBase sequences (from the same miRNA family). Of the 4975 annotated sequences (Table 1) identified in our data set, 33.7% were identified from *Picea abies*, 29.3% from *Pinus taeda* and 11.8% from *Pinus densata* (Table 1). Only 0.8% were identified from *Acacia auriculiformis* and 1.4% from *Zea mays*. Comparing our data with miRBase sequences, we found that 91.7% of annotated *Pinus taeda*, 75% of annotated *Picea abies* and 70% of annotated *Pinus densata* miRNA sequences were also present in the Scots pine sequences. The least represented miRNA database sequences in Scots pine sequences were from *Oryza sativa*—only 13.9% of all *Oryza sativa* sequences present in miRBase were found in Scots pine sequences (Table 2).

**Table 1.** Conserved small RNAs identified from miRBase.

| Annotation | Count | Percentage |
|---|---|---|
| Annotated | 4975 | 0.5% |
| *Acacia auriculiformis* | 41 | 0.8% |
| *Arabidopsis thaliana* | 457 | 9.2% |
| *Oryza sativa* | 307 | 6.2% |
| *Picea abies* | 1676 | 33.7% |
| *Pinus taeda* | 1459 | 29.3% |
| *Pinus densata* | 586 | 11.8% |
| *Populus euphratica* | 2 | 0.0% |
| *Populus trichocarpa* | 162 | 3.3% |
| *Nicotiana tabacum* | 82 | 1.6% |
| *Vitis vinifera* | 132 | 2.7% |
| *Zea mays* | 71 | 1.4% |
| Unannotated | 1,016,721 | 99.5% |
| Total | 1,021,696 | 100.0% |

**Table 2.** Comparison of the number of *P. sylvestris* small RNAs with matching miRNA sequences in miRBase.

| Species | No. of Sequences in miRBase | No. of Matching Sequences | Percentage Found |
|---|---|---|---|
| *Acacia auriculiformis* | 7 | 4 | 57.1% |
| *Arabidopsis thaliana* | 298 | 90 | 30.2% |
| *Oryza sativa* | 592 | 82 | 13.9% |
| *Picea abies* | 40 | 30 | 75.0% |
| *Pinus taeda* | 36 | 33 | 91.7% |
| *Pinus densata* | 30 | 21 | 70.0% |
| *Populus euphratica* | 4 | 1 | 25.0% |
| *Populus trichocarpa* | 352 | 73 | 20.7% |
| *Nicotiana tabacum* | 162 | 30 | 18.5% |
| *Vitis vinifera* | 163 | 27 | 16.6% |
| *Zea mays* | 172 | 37 | 21.5% |

The 4975 annotated Scots pine miRNAs were assigned to 173 miRNA groups (Supplementary File 1), based on mature miRNA sequences from miRBase, creating consensus sequences for the identified miRNAs. Two mismatches were allowed between identified Scots pine miRNAs and annotated plant miRNAs in miRBase. The additional parameters were: additional upstream or downstream bases-2, missing upstream or downstream bases-2. The conifer miRNA sequences reported in various publications, but which were not in miRBase, were compared to miRNA sequences in miRBase in order to assign them to miRNA families, provided that the sequences were publically available. These 173 consensus sequences belonged to a total of 60 conserved miRNA families (Supplementary File 1) The majority of miRNA families (32 families), contained 1 consensus sequence (1 group), while the miR159 family (including miR319) contained the most groups (18). Most of the miRNA families found only in conifers contained only 1 or 2 groups.

Only four conifer species are represented in miRBase v21-*P.taeda* (74 miRNAs), *P.densata* (60 miRNAs), *P.abies* (81 miRNAs) and *C.lanceolata* (9 miRNAs). Comparison of miRBase annotated conifer miRNAs and our data identified 34 conifer specific conserved miRNA families (Table 3). Of these, 34 potentially conifer specific miRNA families, 18 were found in Scots pine, 13 families in loblolly pine and 10 families in Sikang pine, while 24 families were found in Norway spruce. No conifer-specific miRNA families were reported in *C. lanceolata*. Two families—miR950 and miR1311—were identified in four conifer species (*P.sylvestris*, *P.taeda*, *P.densata* and *P.abies*). Fourteen conserved conifer miRNA families were not identified in pine species, but were found only in *Picea abies*. miRNA families miR3699, miR3702 and miR3710 were identified only in *P.sylvestris* and in *Picea abies*, but were not found in the other conifer species represented in miRBase.

It was found that all novel and conserved miRNA families were found in both sample types-control and with MJA treated, but the different isomiRs were also found in different sample types.

**Table 3.** miRNAs found only in conifers based on miRBase and Scots pine data.

| miRNA Family | *P.sylvestris* | *P.taeda* | *P.densata* | *P.abies* | *C.lanceolata* |
|---|---|---|---|---|---|
| miR946 | + | + | + | - | - |
| miR948 | - | + | - | - | - |
| miR947 | + | + | - | + | - |
| miR949 | + | + | - | - | - |
| miR950 | + | + | + | + | - |
| miR952 | + | + | + | - | - |
| miR951 | + | + | - | + | - |
| miR1309 | + | + | - | - | - |
| miR1311 | + | + | + | + | - |
| miR1312 | + | + | + | - | - |

**Table 3.** *Cont.*

| miRNA Family | *P.sylvestris* | *P.taeda* | *P.densata* | *P.abies* | *C.lanceolata* |
|:---:|:---:|:---:|:---:|:---:|:---:|
| miR1313 | + | + | + | - | - |
| miR1314 | + | + | + | - | - |
| miR1315 | + | + | - | - | - |
| miR1316 | + | + | - | - | - |
| miR3693 | - | - | - | + | - |
| miR3694 | - | - | - | + | - |
| miR3695 | - | - | - | + | - |
| miR3696 | - | - | - | + | - |
| miR3697 | - | - | - | + | - |
| miR3698 | - | - | - | + | - |
| miR3699 | + | - | - | + | - |
| miR3700 | - | - | - | + | - |
| miR3701 | + | - | + | + | - |
| miR3702 | + | - | - | + | - |
| miR3703 | - | - | - | + | - |
| miR3704 | - | - | + | + | - |
| miR3705 | - | - | - | + | - |
| miR3706 | - | - | - | + | - |
| miR3707 | - | - | - | + | - |
| miR3708 | - | - | - | + | - |
| miR3709 | - | - | - | + | - |
| miR3710 | + | - | - | + | - |
| miR3711 | - | - | - | + | - |
| miR3712 | + | - | + | + | - |

### 3.3. Identification of Potential miRNA Precursors

The determination of potential novel miRNAs by identification of miRNA precursors, as well as the identification of precursors for conserved miRNAs, was performed using the miRA tool by mapping the mature miRNA sequences to the *Pinus* PGI_v9.0_032811 unigene sequences. Additional *P. sylvestris* sequence databases were also analyzed (e.g., expressed *P. sylvestris* sequences in GenBank and the draft *P. sylvestris* genome); however, these yielded a smaller number of potential precursor sequences in comparison to the *P. sylvestris* unigene set. Therefore, only this database was subsequently utilised.

A total of 1029 potential novel miRNAs that had no homology (as described previously) to miRBase v22 annotations were found. They were grouped into 34 families and 46 predicted precursor sequences were identified (Supplementary File 2). The largest family was miR00005 and contains 124 miRNA isoforms; the smallest family was miR00004, with 3 isoforms (Supplementary File 2). Most of novel miRNAs were located on the 3'arm (526 sequences), 216 novel miRNAs were located on the 5'arm, and 287 miRNA sequences were identified as star sequences.

Analyzing the 4975 Scots pine mature miRNA sequences with homology to miRBase miRNAs, 50 potential precursor sequences for 2209 of these mature miRNAs were identified. These 50 sequences were predicted to be precursors of 20 families (Supplementary File 3). The most isomiRs (483), were found for miR950, but the most precursors were found for miR482 and miR950.

Using the miRA tool, only 19 potential precursor sequences for 780 of these mature miRNAs were identified (Supplementary File 4). These 19 sequences were predicted to be precursors of 9 miRNA families-miR396, miR482, miR946, miR949, miR952, miR1312, miR1313, miR1314 and miR3701. All 19 families were also found using manual searching and selection.

A large number of isomiRs that were homologous to each of these families were identified with the miRA tool, ranging from 242 isomiRs for the miR482 family to 23 isomiRs for the miR946 family. Only one potential precursor sequence was found for the miRNA families (miR946, miR952, miR1312, miR1313), while five potential precursor sequences were found for the family miR482. Of the 780 isomiRs, 342 were found on the 3' arm, 250 on the 5' arm, but 188 isomiRs were identified as star

sequences. Only one precursor identified with the miRA tool (TC159053 for miR952) was not identified by manual searching and selection.

In addition to the miRA tool, potential precursor sequences for conserved miRNAs were analyzed for the ability to form the required stem-loop structure, and the minimal folding free energy indexes were calculated (Supplementary File 3). Predicted pre-miRNA sequences were trimmed in the primary miRNA sequence region until the next bulge or loop after the miRNA* region. Minimum folding free energy indexes ranged from 0.72–1.31, with most being > 0.85 (47 precursor sequences (94%)) (Supplementary File 3), and corroborated with previously reported sequences. Three predicted precursors (for more than 250 isomiRs) had an MFEI less than 0.85. A number of sequences were identified as potential precursors for multiple miRNAs which are in the same family. This is due to the parameters utilized, which allows a maximum of two mismatches between a miRNA and a potential precursor miRNA sequence. Two or more than two potential precursor sequences were identified for miR396, miR482, miR949, miR1314, miR3701. In some cases, the bioinformatic identification of potential precursor sequences was complicated, because the mature miRNA was identical to the sequence in one precursor, but the MFEI was lower than in a different potential precursor with less homology to the mature miRNA sequence. These mismatches between potential precursors and mature miRNAs, as well as between two potential precursors could be due to the sequences being derived from different individuals. Another possibility is that either the true precursor or mature miRNA sequences are not present in the data set or databases.

*3.4. miRNA Target Identification*

Analysis of conserved miRNA putative target genes identified 119 genes targeted by 58 miRNA families (Supplementary File 3). Seven of these target genes, TC195763, TC162935, DR694512, CF470498, TC170132, TC159334, and TC195763, were targets of two different miRNA families. Ten of the putative gene targets (targeted by eight miRNA families) were of unknown function. Of these eight families, four miRNA families targeted one or more unannotated genes, but four families targeted multiple genes with both unknown and known functions. No putative targets were identified for seven miRNA families (defined as "no result"). However, putative target genes for six of these miRNA families have been described in other publications [53–58].

Six of the identified putative target genes (targeted by five miRNA families—miR393, miR950, miR951, miR3623, and miR024) were homologous to *TIR/TIR like/NBS-LRR* disease-resistance protein genes and an additional three targets (targeted by three miRNA families) were with unknown function, but these miRNAs have been previously described as targeting disease resistance protein genes. Plant NBS-LRR proteins are involved in the detection of pathogen-associated proteins, most often the effector molecules of pathogens which are responsible for virulence.

GO annotation [52] of the identified target genes indicated that the most common functions in the biological process domain (Supplementary File 5, Figure S1) were related to transcription regulation and signal transduction, as well as protein phosphorylation and ubiquitination, and response to stress. The most common GO annotations in the molecular function domain (Supplementary File 5, Figure S2) were binding, including DNA binding, and transcription factor activity.

Analysis of novel miRNA putative target genes identified 136 genes targeted by 28 families (from a total of 34 potential novel miRNA families identified by precursor sequence analysis) (Supplementary File 2) or 217 isoforms (from 1029 miRNA isoforms) with 471 target sites. These results indicated that the same isoforms can target not only different target genes, but also more than one target site within a single target gene, and also that different isoforms from more than one family can target the same target gene (sequences with red color in Supplementary File 3). The largest number of target genes were predicted for family miR00002 with 20 targets. The most target sites (65) were found for family miR00025, where 38 isoforms targeted three target genes and family miR00027, where 15 isoforms targeted three target genes and 58 target sites. Fourteen miRNA families were homologous to resistance-related target genes. Four potential target genes (targeted by five miRNA

families or 19 isoforms) were of unknown function. Sixteen target genes were homologous to alcohol dehydrogenases, eight target genes were homologous to phytocyanins and seven were homologous to peroxidases. Using the Swissprot protein database, the most common GO annotations of the identified target genes in the biological process domain (Supplementary File 5, Figures S3 and S4) were response to stimulus (including response to abscisic acid, response to cytokinin, heat acclimation, response to wounding, hypoxia and others), single organisms process and cellular process. The most common GO annotations in the molecular function domain were binding and catalytic activity (Supplementary File 5, Figure S5). Utilizing the larger to non-redundant protein database, the most common GO annotations of the identified target genes in the biological process domain (Supplementary File 5, Figure S6) were metabolic processes, single organism processes, cellular processes and responses to stimuli. However, the most common GO annotations in the molecular function domain were the same-binding and catalytic activity (Supplementary File 5, Figure S7).

## 4. Discussion

Potential novel miRNAs were determined by identifying precursor sequences and target genes. Most novel isomiRs were located on the 3′ arm of the precursor stem-loop structure, similarly to the conserved miRNA sequences, and corresponding to previous reports [59]. miRNA isoforms can be processed not only by the Dicer enzyme, but also by other Dicer-like enzymes and Dicer independent mechanisms. This suggests that only a subset of small RNAs can be identified as mature miRNAs in concordance with a Dicer mediated miRNA biogensis pathway, but other isoforms might be produced by other Dicer-like enzymes or by Dicer-independent mechanisms, or can be caused by recurrent somatic mutations in Drosha, which can induce changes in miRNA expression, and somatic mutations in Drosha and Dicer1 can impair miRNA biogenesis [60,61].

It has been reported that more than 90% of miRNA precursors had an MFEI greater than 0.85, and no mRNAs, tRNAs, or rRNAs had more MFEI higher than 0.85 [62]. Our data indicate that most minimum folding free energy indexes were > 0.85 and included 94% of all precursors, in corroboration with previously reported data.

Comparing our data, we concluded that only 23 from 58 miRNAs—including isomiRs, from *Taxus chinesis* [29], which also had treatment with MJA—were also found in our data, from which more than half (15) of the miRNA sequences were found in the same sample (control or MJA-treated) type as in *T. chinesis*. In some other gymnosperm miRNA studies [25,28,63] more miRNA families were found that have precursors, but in this study, more isomiRs were found for these families, compared to the previous three studies. These differences in identified isomiR number between studies is probably due to the utilized methodology and parameters.

A survey of miRNA and other small RNA sequences across a wide range of species has identified conserved miRNA families [64], and the miRNA families identified from our study is broadly consistent with this report. The five miRNA families classified as ubiquitous (miR156, miR166, miR167, miR168, miR172) were all identified in our data. In addition, the 16 miRNA families classified as present in most taxonomic groups were also present in our data, with the exception of miR4414. None of the miRNA families enriched in dicots were found in our data, and they were also poorly represented in other gymnosperm species (*Cycas rumphi*, *Ginkgo biloba* and *Picea abies*) [64]. All three miRNA families reported to be enriched in gymnosperms (miR536, miR1083, miR1314) were identified in our dataset. Of the 20 miRNA families enriched in monocots, only one (miR1432) was identified in this study. All of the miRNA families conserved in *Arabidopsis*, *Oryza* and *Populus* [65], were also identified within the sequences expressed in *Pinus sylvestris*. However, the miRNAs that were enriched in monocots or dicots were not identified in *P.sylvestris* [64]. A number of putative gymnosperm-specific miRNAs were identified; however, in many cases, their sequences were less conserved than the highly conserved plant lineage miRNAs identified. For example, the most highly conserved miRNA sequences were found between *P. sylvestris* and *A. thaliana*, while the miRNA families that were common with other gymnosperm species had one or more nucleotide mismatches compared to the *P. sylvestris* sequences.

Many of the highly conserved miRNAs are found in a range of plant groups (e.g., monocots, dicots and gymnosperms), suggesting a common ancestry and function for these miRNAs. However, some of the *P. sylvestris* miRNA families were similar to those miRNAs only reported to be present in monocots (i.e., miR1432, miR2275, miR5072, miR5083). In some cases, this could be due to the underrepresentation of a particular miRNA family in miRBase (e.g., miR5072 or miR5083—which are only reported in rice). However, in other cases (miR1432, miR2275) these families are reported to be present in a range of monocot species, including rice, sorghum, maize, *Brachypodium* and *Aegilops*). These miRNA families were also reported at least one other gymnosperm species, not only in monocots. Two conserved miRNA families identified in *P. sylvestris* (miR6024, miR6478) have not been previously reported in any other gymnosperm species.

The miRNA families miR156, miR159/319, miR160, miR166, miR171, and miR408 are reported to be present in all green plants (embryophyta); the miR396 family is present in all vascular plants (tracheophytes); while miR397 and miR398 are present in all seed plants (spermatophytes) [66]. The miR403, miR828 and miR2111 families were reported as eudicot-specific families; however, these miRNA families, except for miR403 (restricted to core eudicot lineages), were also identified in *P.sylvestris* in this study. Only some families-miR156, miR159/319, miR160, miR166, miR171, miR408, miR390/391, miR395, miR529 and miR536-identified in bryophytes were also found in *P.sylvestris*, which could be a result of the relatively small number of miRNAs reported in bryophytes.

The higher proportion of perfect matches with angiosperms rather than conifers may be due to the low number of conifer sequences compared to angiosperms and the lack of conserved conifer miRNA sequences in the database, or it may indicate that miRNA sequences are more diverse in conifers than in angiosperms. Conserved miRNA families are relatively easy to identify, not only because they are annotated with a higher confidence from a range of species, but also because they are reported to be more abundant, with the 21 most highly conserved miRNA families accounting for 54–98% of miRNA sequences [64]. Therefore, species or lineage-specific miRNA families may be unrecognized due to inadequate sequence coverage in addition to the difficulties in unambiguously identifying novel miRNAs.

## 4.1. Comparison of Scots Pine miRNA Families with Published Conifer miRNAs

A literature review identified 22 publications reporting on miRNA studies in conifers, of which seven publications were about studies in five different pine species—*P.taeda*, *P.densata*, *P.strobus*, *P.contorta*, *P.tabuliformis*. Of these, miRNA sequences from only four publications are available in miRbase: Lu et al. [33] (*Pinus taeda*), Wan et al. [67] (*Pinus densata*), Yakovlev et al. [26] (*Picea abies*) and Wan et al. [28] (*Cunninghamia lanceolata*). Comparing our data with previously reported miRNA families, we found that of the 58 conserved miRNA families identified in Scots pine, 56 also were reported in at least one other conifer species (Supplementary File 2). Eighteen miRNA families were found only in conifer species (*Pinus taeda*, *Pinus densata*, *Picea abies*): miR950, miR951, miR952, miR946, miR947, miR949, miR3699, miR3701, miR3702, miR3710, miR3712, miR1309, miR1311, miR1312, miR1313, miR1314, miR1315, and miR1316. The miR6024 and miR6478 families were identified only in Scots pine, but two families, miR5072 and miR5083, were found in Scots pine and in only one additional conifer species. The functional annotation of the target genes of these conserved miRNA families indicates that most are involved in plant growth and development, biotic and abiotic stress response and disease resistance.

## 4.2. Target Genes

The majority of highly conserved miRNA families (e.g., as reported in Cuperus et al. [66]), which are found in all plants including bryophytes, target transcription factors such as SPL, MYB, HD ZIP III, AP2 and others [68]. In contrast, none of the gymnosperm-specific miRNA target genes were annotated as having DNA-binding functions, suggesting that transcription factors are underrepresented in this group. However, this could also be an artefact due to the small number of gymnosperm-specific

miRNAs and target genes. In addition, transcription factors may be underrepresented in the publicly available *P. sylvestris* gene sequences. In this study, the predicted target genes included stress-related gene families such as Leucine Rich Repeat (*LRR*) protein genes, protein kinase domain, Toll-Interleukin receptor (TIR) domain, disease resistance protein RPP1-WsB, heat shock proteins, and blue cooper proteins (including uclacyanin, plantacyanin). Transcription factors were also among the identified target genes-CCAAT-binding transcription factors, AP2-related transcription factors, homocysteine S-methyltransferases, MYB-like proteins, regulatory protein GntR, *MarR* family genes, *GAMYB*, and zinc finger protein genes. Target genes also included transferases and other enzymes. A similar functional distribution of target genes was also reported in crop species, as almost 50% of the miRNA targets were transcription factors in pathways that are likely important in setting or maintaining the developmental program leading to high quality soybean seeds [69], and 66% of target genes in crops are reported to be transcription factors; however, 11% are lncRNA, 7% are NB-LRR proteins, 2% are pathogen proteins and 24% are other proteins [70].

### 4.3. Resistance/Stress Genes

*TIR-NBS-LRR* and other *NBS-LRR* genes are involved in defense responses to pathogen infection and disease resistance. This is a highly diverse class of genes that can include highly conserved genes as well as lineage- or species-specific genes. They can form gene clusters, and are regulated by miRNAs [71]. The *TIR-NBS-LRR* gene family has been lost in monocots [72], but is present in gymnosperms. The conserved miR482/2118 family targets the highly-conserved P-loop motif in *NBS* genes [71], and was also identified in this study. Three main mechanisms of miRNA locus evolution have been proposed: the inverted duplication of a target gene sequence, leading to the formation of the required stem-loop structure [73]; the formation of a stem-loop structure by the self-complementarity of a transcribed sequence [74]; the mutation of existing miRNA sequences [75]. Zhang et al. [71] suggest that the clustering of *NBS-LRR* genes facilitates the expression of inverted tandem duplications of target genes, thereby facilitating the tandem co-evolution of miRNAs and their target *NBS-LRR* genes. Of the gymnosperm/conifer specific miRNA families identified in this study, at least two target *NBS-LRR* genes (miR1311 and miR1312) and the targets of the majority of the others are resistance genes. Conifer genomes are reported to contain a larger proportion of gene families in comparison to angiosperms [76], and contain a large proportion of repeated sequences and retrotransposons [77,78] which can facilitate the formation of inverted genes and other sequences. Further characterization of novel miRNAs in Scot pines will reveal if these unique miRNAs share similar origins.

### 4.4. Transcription Factors

Many genes are activated in response to stresses at the transcriptional level, and they provide stress tolerance through the production of vital metabolic proteins and also by regulating downstream genes [79]. Transcription factors (TFs) are essential for the regulation of gene expression, and usually belong to members of multigene families [80]. TFs families can evolve by a range of mechanisms such as exon capture, duplication, translocation and mutation [81,82]. The majority of miRNA targets are transcription factors, which regulate plant growth and developments [83–85]. The role of miRNAs in the regulation of transcription factors influencing traits such as meristem identity, polarity and flowering has been established; however, even highly conserved miRNA families can be involved in different developmental processes, and their role and mechanisms of action can vary between species, or even between different tissues types [86]. Therefore, the role of these conserved miRNAs must be further investigated in conifers, as these miRNA families may have evolved to play different roles to model angiosperm species.

### 5. Conclusions

A large number of mature and precursor miRNA sequences have been identified and published in miRBase and other databases, but there are still problems with the correct identification and annotation

of miRNAs and the subsequent accuracy of available information [87,88]. The consequences of this are inconsistencies in the naming of miRNA families and isoforms, leading to difficulties in comparisons between studies. It is also difficult to identify true precursor miRNAs based on homology searches with mature miRNA sequences. There are several tools available for the prediction (MiRFinder, Rfam, MIReNA, etc.) of precursor miRNAs, but distinguishing true pre-miRNAs from other hairpin sequences with stem loop structures can be complicated, as these structures are not unique to pre-miRNAs, and many other coding or non-coding RNAs—such as rRNAs, tRNAs and mRNAs—can also form similar hairpin structures.

The majority of highly conserved plant miRNAs were identified in this study, as well as some conserved miRNAs previously reported to be monocot specific. No conserved dicot-specific miRNAs were identified. A number of potential gymnosperm or conifer specific miRNAs were found, shared among a range of conifer species. Potential target genes were identified, of which the targets of highly conserved miRNAs present in most plant families were transcription factors, while the conserved conifer-specific miRNA targets were involved in disease resistance.

**Supplementary Materials:** The following are available online at http://www.mdpi.com/1999-4907/11/4/384/s1, Supplementary File 1: Table S1: Conserved miRNAs grouped in groups by consensus sequence, Supplementary File 2: Novel miRNAs, their targets, families and Blast2Go data, Supplementary File 3: Excel file S1: conserved miRNA precursors, targets, Supplementary File 4: Excel file S1: identified conserved miRNA precursors by miRA tool. Supplementary File 5: Figure S1: Conserved miRNA target sequences distribution in category of biological process, Figure S2: Conserved miRNA target sequences distribution in category of molecular function, Figure S3: Novel miRNA target sequences distribution in category of biological process in SwissProt database, Figure S4: Novel miRNA target sequences distribution in category in response to stimulus in SwissProt database, Figure S5: Novel miRNA target sequences distribution in category of molecular function in SwissProt database, Figure S6: Novel miRNA target sequences distribution in category of biological process in non-redundant protein database, Figure S7: Novel miRNA target sequences distribution in category of biological process in non-redundant protein database.

**Author Contributions:** B.K., D.R., C.G.F. conceived and designed the research, I.Š. cultivated, treated and provided the scots pine ramets, B.K., I.Š. and V.Š. performed the experiments, B.K. and D.R. analyzed and investigated the data and wrote the manuscript (original draft paper), A.V.-S., I.Y. and C.G.F. trained, consulted and assisted with sequencing with the IonTorrent PGM. All authors reviewed and edited the manuscript and approve the published version of the manuscript.

**Funding:** This project was funded by the Latvian Council of Science grant 284/2012 "Investigation of molecular defense mechanisms in Scots pine (*Pinus sylvestris* L.).

**Conflicts of Interest:** The authors declare no completing interests.

**Data Availability:** Unique small RNA sequences were deposited to the SRA (Short Read Archive, NCBI) with the accession number PRJNA531446 (https://www.ncbi.nlm.nih.gov/sra/?term=PRJNA531446).

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
