# Peer review of "Identification and in Silico Characterization of Novel and Conserved MicroRNAs in Methyl Jasmonate-Stimulated Scots Pine (Pinus sylvestris L.) Needles"

_forests, doi:10.3390/f11040384_

Round 1
Reviewer 1 Report
The article by Krivmane et al. concerns with the identification and characterization of miRNA from Scots pine. The article is well written and the results are well described.
The article could be improved by providing a comparision between miRNA from "non-elicitated" trees and the ones treated with MeJA, in order to better clarify to the reader which miRNA are overespressed during MeJA treatment.
Moreover, it would be possible to quantify, i.e. by quantitative PCR, the amont of the common miRNA?
Author Response
Point 1: The article could be improved by providing a comparision between miRNA from "non-elicitated" trees and the ones treated with MeJA, in order to better clarify to the reader which miRNA are overespressed during MeJA treatment.
Response 1: In plants, miRNA studies have been mainly concentrated in angiosperms and relatively few reports have been made in conifers. There are no reports about miRNAs in Scots pine, therefore this article was initially planned to concentrate only on identification and in silico characterization of novel and conserved microRNAs. We are working currently on the second manuscript based on these data about miRNAs and as well as transcriptome expression changes from "non-elicitated" trees and the ones treated with MeJ from the same study. In the second manuscript we will discussed in more detail about up-regulated, down-regulated miRNAs, isoforms and target genes after treatment with MeJ.
Point 2: Moreover, it would be possible to quantify, i.e. by quantitative PCR, the amont of the common miRNA?
Response 2: Real-time RT-PCR technologies have been developed to amplify and quantify mature and the precursor miRNA (pre-miRNA). There are different techniques that include use of stem–loop reverse transcriptase–PCR, polyadenylation of RNAs, ligation of adapters or RT with complex primers, using universal or miRNA specific qPCR primers or probes. Many of these methods are designed for the expression analysis of mature miRNAs or for the study of pre-miRNAs, or pri-miRNAs. Due to the short length of mature miRNAs and the high similarity of multiple members of miRNA families, expression analysis of miRNAs by qPCR has several particular challenges.
Isoforms present another problem, which needs to be carefully considered when designing primers to quantify miRNA by real-time PCR. Numerous miRNAs exists as isoforms of nearly identical mature and precursor sequences. Using SYBR green detection, it is often not possible for the PCR primers designed to the hairpin to discriminate among the various isoforms. This issue can be resolved with TaqMan™ probes that anneal to the loop portion of the precursor. It is not practical to purchase dozens of TaqMan™ probes to all the miRNA. Another problem is, that there is not available high quality Pinus Silvestris reference gene sequence, that make a problem for successful discovery of Scots pine precursors and for successful primer design for quantitative PCR with real-time PCR.
It is not possible to quantify, i.e. by quantitative PCR, the amount of the common miRNA for all miRNA families and isoforms together at one real-time PCR reaction.
Reviewer 2 Report
So far, methyl jasmonate induced gene expression in Scots pine has not been extensively studied. This important molecule in the signaling pathway especially during plant defense process turned out to be also involved in expression of 136 potential target genes and this via 1029 potential novel miRNAs in Pinus sylvestris seedlings. Despite the difficulty in identification of conserved specific miRNAs present in dicotyledons, the paper has a great scientific value. The targeted genes in response to Met-JA are not only involved in plant response to the pathogen attack, but also in general mechanisms involving the transcription factors playing role in many cellular processes. The majority of miRNA targets are transcription factors, which regulate plant growth and developments. Summing-up, the paper clearly demonstrated the unrevealed role of the jasmonates in Scots pine response to biotic and abiotic stresses. I recommend publishing the article after very minor corrections.
About the content:
The article is original, has good technical quality and large general interest.
The title of the article and keywords clearly reflect paper's content.
Introduction presents the problem clearly.
Experimental methods are adequate and well described. A high throughput sequencing and computational prediction were appropriately used in order to predict the conserved miRNAs molecules.
Results are fully presented, supported by supplementary tables and figures.
Discussion is largely devoted to the conserved miRNA families in different plant species in regard to Scots pine obtained data. Difficulty in clear discernment of novel conserved miRNA in Scots pine was well explained.
References are complete [except position no 63] and adequate.
About Presentation:
Length is commensurate with the paper's content.
Quality of tables and figures is adequate and supplementary files of great usefulness.
The English language is adequate.
About Scientific evaluation:
The general scientific approach is properly stated and well explained.
Some very little errors occurred:
Line 4: The abbreviation of Linnaeus should be written “L.”
Line 223: Latin name of P. sylvestris should be italicized
Line 403: Scots “pine” not “Pine”
In Discussion the reference [63] is missing after the [62] mentioned
Line 566 and 569: Larix leptolepis should be italicized,
as well as other Latin names of species mentioned in References, e.g. Cunninghamia lanceolata, Taxus chinensis, Pinus taeda, etc.
Author Response
Point 1. Some very little errors occurred:
Line 4: The abbreviation of Linnaeus should be written “L.”
Line 223: Latin name of P. sylvestris should be italicized.
Line 403: Scots “pine” not “Pine”.
In Discussion the reference [63] is missing after the [62] mentioned.
Line 566 and 569: Larix leptolepis should be italicized, as well as other Latin names of species mentioned in References, e.g. Cunninghamia lanceolata, Taxus chinensis, Pinus taeda, etc.
Response 1:
There were lost the formatting (italic style) after coping the manuscript text in the template, therefore we have done some corrections in Latin and gene names. All corrections are included in the manuscript text using track changes now.
Line 4: The abbreviation of Linnaeus has been changed to “L.”.
Line 77: The name of in vitro has been italicized.
Line 233: Latin name of P. sylvestris has been italicized.
Line 355: In discussion the wrong mentioned reference [64] has been changed to [63].
Line 403: The name Scots “Pine” changed to “pine”.
Line 534: The reference 11th has been transfered from line 534 to line 535.
Line 550: Latin names of Pinus contorta and Oryza sativa has been italicized.
Line 553: Latin name of Pinus strobus and the name of in vitro has been italicized.
Line 557: Latin name of Larix leptolepis has been italicized.
Line 563: Latin name of Larix leptolepis has been italicized.
Line 566: Latin name of Larix leptolepis has been italicized.
Line 569: Latin name of Larix leptolepis has been italicized.
Line 586: Latin name of Cunninghamia lanceolata has been italicized.
Line 592: Latin name of Taxus chinensis has been italicized.
Line 594: Latin name of Sequoia sempervirens has been italicized.
Line 597: Latin name of Pinus taeda has been italicized.
Line 601: Latin name of Pinus taeda has been italicized.
Line 603: Latin name of Pinus taeda has been italicized.
Line 607: Latin name of Arabidopsis thaliana has been italicized.
Line 612: Latin name of Nicotiana attenuata has been italicized.
Line 622: Latin name of Nicotiana attenuata has been italicized.
Line 626: Latin name of Taxus × media has been italicized.
Line 634: Latin name of In silico has been italicized.
Line 635: Latin name of Arundo donax has been italicized.
Line 659: Latin name of Arabidopsis has been italicized.
Line 665: Latin name of Arabidopsis has been italicized.
Line 669: Latin name of Arabidopsis has been italicized.
Line 691: Latin name of Pinus densata has been italicized.
Line 707: The gene name NBS-LRR has been italicized.
Line 713: Latin name of Arabidopsis thaliana has been italicized.
Line 715: Latin name of Arabidopsis thaliana has been italicized.
Line 729: Latin name of Phaseolus vulgaris has been italicized.
Line 735: Latin name of Marchantia polymorpha has been italicized.
Line 741: Latin name of Medicago sativa has been italicized.